# Influence of Fluid Viscosity and Compressibility on Nonlinearities in Generalized Aerodynamic Forces for T-Tail Flutter

Dominik Schäfer 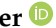

Institute of Aeroelasticity, German Aerospace Center (DLR), Bunsenstraße 10, 37073 Göttingen, Germany; d.schaefer@dlr.de

**Abstract:** The numerical assessment of T-tail flutter requires a nonlinear description of the structural deformations when the unsteady aerodynamic forces comprise terms from lifting surface roll motion. For linear flutter, a linear deformation description of the vertical tail plane (VTP) out-of-plane bending results in a spurious stiffening proportional to the steady lift forces, which is corrected by incorporating second-order deformation terms in the equations of motion. While the effect of these nonlinear deformation components on the stiffness of the VTP out-of-plane bending mode shape is known from the literature, their impact on the aerodynamic coupling terms involved in T-tail flutter has not been studied so far, especially regarding amplitude-dependent characteristics. This term affects numerical results targeting common flutter analysis, as well as the study of amplitude-dependent dynamic aeroelastic stability phenomena, e.g., Limit Cycle Oscillations (LCOs). As LCOs might occur below the linear flutter boundary, fundamental knowledge about the structural and aerodynamic nonlinearities occurring in the dynamical system is essential. This paper gives an insight into the aerodynamic nonlinearities for representative structural deformations usually encountered in T-tail flutter mechanisms using a CFD approach in the time domain. It further outlines the impact of geometrically nonlinear deformations on the aerodynamic nonlinearities. For this, the horizontal tail plane (HTP) is considered in isolated form to exclude aerodynamic interference effects from the studies and subjected to rigid body roll and yaw motion as an approximation to the structural mode shapes. The complexity of the aerodynamics is increased successively from subsonic inviscid flow to transonic viscous flow. At a subsonic Mach number, a distinct aerodynamic nonlinearity in stiffness and damping in the aerodynamic coupling term HTP roll on yaw is shown. Geometric nonlinearities result in an almost entire cancellation of the stiffness nonlinearity and an increase in damping nonlinearity. The viscous forces result in a stiffness offset with respect to the inviscid results, but do not alter the observed nonlinearities, as well as the impact of geometric nonlinearities. At a transonic Mach number, the aerodynamic stiffness nonlinearity is amplified further and the damping nonlinearity is reduced considerably. Here, the geometrically nonlinear motion description reduces the aerodynamic stiffness nonlinearity as well, but does not cancel it.

**Keywords:** aeroelasticity; T-tail flutter; quadratic mode shape components; CFD unsteady aerodynamics; nonlinear dynamics

## 1. Introduction

The assessment of T-tail flutter requires unsteady aerodynamic forces beyond the scope of the conventional Doublet Lattice Method (DLM), usually accounted for by means of correctional terms computed by external methods and superposed with the DLM aerodynamics. Common approaches involve computing unsteady aerodynamic forces due to lifting surface roll and in-plane motion by means of a strip theory method [1–3]. The theory is based on tilting the steady aerodynamic forces in phase to the lifting surface roll motion in addition to computing aerodynamic forces due to in-plane motion with the methods presented in [4]. The phase lag between structural displacements and aerodynamic response

is accounted for by Theodorsen's lift deficiency function [5]. L. van Zyl develops a more integrated way of calculating generalized aerodynamic forces for T-tail flutter assessment by using an extended DLM algorithm in [6,7]. This approach takes into account steady aerodynamic forces and all relevant physical degrees of freedom on the aerodynamic box level. An additional discretization of the geometry for the aerodynamic correction method, hence, is not necessary. A comprehensive survey of the methods for augmenting unsteady aerodynamic forces from conventional approaches for T-tail flutter assessment in addition to the application of an unsteady vortex lattice method is presented in [8]. Alternatively, unsteady CFD methods may be used to inherently capture the aerodynamic forces to their full extent. Within the scope of the development of a tool for the estimation of flutter boundaries at transonic flight speeds based on linear structure and nonlinear, inviscid aerodynamics, numerical studies and wind tunnel experiments are compared in [9]. The wind tunnel model used for the verification, however, features a very stiff Vertical Tail Plane (VTP) and, therefore, does not show the common T-tail flutter phenomenon, usually consisting of VTP out-of-plane bending and torsion. Application of a flutter assessment process for a free-flying aircraft with T-tail incorporating CFD aerodynamics is shown in [10], including the solution of the trim load and static deformation. The procedure is applied to the Piaggio P180 aircraft and is based on the premises that the nonlinear steady state flowfield has a significant impact on the flutter stability, but the response of the flowfield to small disturbances can be considered linear. An iterative scheme is applied to align the trimmed aircraft with the linearized system. Isogai [11,12] refers to the experiments presented in [13] and emphasizes the need for methods capable of predicting transonic T-tail flutter due to the unusually sharp transonic dip that was shown by the experiments. In his work, he illustrates the development and application of a 3D Navier–Stokes code especially designed to include the aerodynamic forces due to lifting surface in-plane motion. The code is applied to T-tail configurations without and with swept and tapered vertical and horizontal tail planes in transonic flow conditions. Santos [14] addresses the development of a framework for numerical flutter assessment incorporating CFD aerodynamics for industrial applications. The benefit of a comprehensive aerodynamic method especially for the transonic flow condition is shown for numerical flutter studies of a T-tail wind tunnel model. However, both the uncoupled, as well as the coupled fluid–structure interaction routines are based on a linear modal approach, and this drawback is emphasized in the outlook of the paper.

While the proper description of unsteady aerodynamic forces for T-tail flutter has been the focus of the research community since the fatal crash of the Handley Page "Victor" bomber in 1954 [15], the literature has also shown that the full description of the unsteady aerodynamic terms in combination with a linear modal approach for the representation of the dynamical system may lead to spurious stiffness terms [16]. For a physically more accurate flutter assessment of T-tails, it is suggested to include quadratic deformation components in the modal representation at least of the VTP out-of-plane bending. These additional deformation components are usually obtained from linear [16,17] or nonlinear finite element analyses [18,19]. This extended modal formulation is also used in aeroelastic problems involving highly flexible wings, e.g., [19–21].

The effect of the quadratic mode shape components on the stiffness of the VTP out-of-plane bending mode shape is known from the literature [8,16], but their impact on the aerodynamic coupling terms has not been studied yet, especially regarding amplitude-dependent characteristics. This would affect numerical results targeting common flutter analysis, as well as the study of amplitude-dependent dynamic aeroelastic stability phenomena, e.g., Limit Cycle Oscillations (LCOs). In particular, the VTP out-of-plane bending deformation, which results in a Horizontal Tail Plane (HTP) roll motion, induces an aerodynamic yaw moment, which performs mechanical work on the VTP torsion (HTP yaw motion). These aerodynamic work terms have recently been shown to be nonlinear with respect to the displacement amplitude at subsonic Mach numbers and, additionally, change significantly when higher-order displacement terms are included in the numerical stud-

ies [22]. Here, the aerodynamic nonlinearity is studied using inviscid flow at a Mach number of 0.4 and is shown to be sensitive to drag forces, which raises the assumption that it might as well be susceptible to viscous aerodynamic forces. Furthermore, as future transport aircraft will still operate at transonic flow conditions, the effect of fluid compressibility on the aerodynamic nonlinearity needs to be addressed. This paper will take up the results presented in [22] and focus on a comparison between the nonlinear generalized aerodynamic responses in inviscid and viscous flow with regard to the frequency of oscillation, the deformation amplitude, and the linearity of the deformation. In addition, the results for a transonic Mach number and viscous flow conditions will be presented. With these studies, the knowledge about aerodynamic nonlinearities occurring in T-tail flutter and their dependencies on geometric structural nonlinearity is expanded and the proper assessment of dynamic aeroelastic instabilities is supported. Although the primary objective is the investigation of the aerodynamic coupling term between HTP roll and yaw motion for T-tail configurations, the results may be transferable to other configurations with intersecting lifting surfaces, e.g., H-tails, U-tails, or slender wings with winglets.

The paper is structured as follows: Section 2 will outline the details of the approach selected to address the research question. This covers a description of the simulation models, as well as the forced motion procedure and the way in which the results are assessed. Section 3 follows with a presentation of the results, focusing first on inviscid subsonic flow and advancing towards viscous transonic flow. The results are discussed in Section 4. Here, the identified aerodynamic nonlinearities are studied regarding their physical sources and put into perspective regarding T-tail flutter. Section 5 summarizes the findings and proposes the next steps to be taken for further studies.

## 2. Approach and Simulation Models

To study the effect of viscosity and compressibility on the aerodynamic response to structural deformation, the focus is set on an isolated HTP derived from a generic T-tail configuration described in [8,23]. This facilitates studying the aerodynamic response and its dependencies on the displacement amplitude, fluid viscosity, and fluid compressibility without aerodynamic interference effects. The HTP, illustrated in Figure 1, has a span of 8 m, a constant chord length of 2 m, is unswept, and without a dihedral. The airfoil is a symmetric NACA 0012. Previous studies on the generic T-tail have revealed a minimum flutter speed at an incidence angle of 3.0° [23], for which reason this incidence angle was chosen for the presented studies. With a reference surface area of 16 m² and the reference values as listed in Table 1, this setting results in an up force and a positive lift coefficient of roughly 0.208 at Mach 0.4 and 0.259 at Mach 0.8.

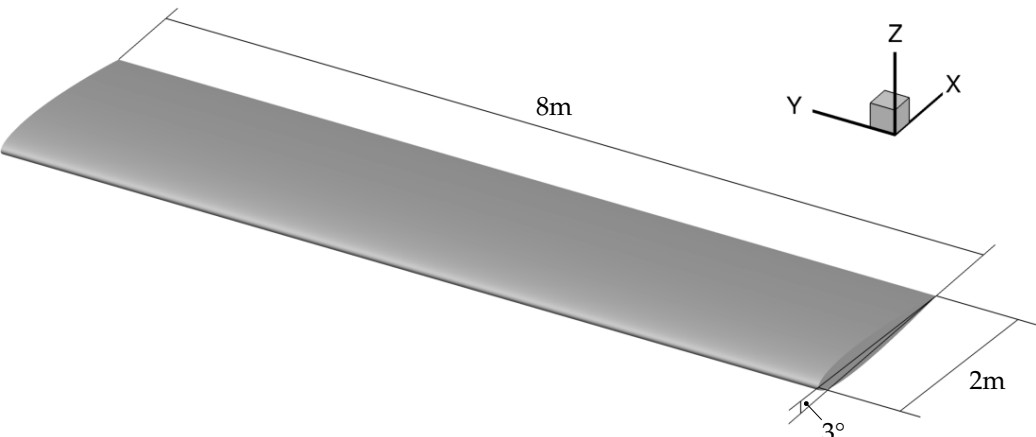

**Figure 1.** Geometry of the isolated HTP.

**Table 1.** Simulation parameters.

| Parameter | Symbol | Value |
|---|---|---|
| Mach numbers/- | $Ma$ | 0.4 │ 0.8 |
| Reduced frequencies/- | $k$ | 0.056, 0.231 |
| Frequencies/Hz | $f$ | 1.213, 5.004 [†] │ 2.426, 10.008 [‡] |
| Rotational amplitudes/° | $\varphi \mid \psi$ | 0.01, 0.917, 1.834, 3.669, 5.000 |
| Temperature/K | $T$ | 288.15 |
| Density/kg m$^{-3}$ | $\rho$ | 1.225 |
| Dynamic viscosity/N s m$^{-2}$ | $\nu$ | $17.89 \times 10^{-6}$ |
| Ratio of specific heats/- | $\kappa$ | 1.4 |
| Ideal gas constant/J kg$^{-1}$ K$^{-1}$ | $R$ | 287 |
| Reduced frequency reference length/m | $\bar{c}$ | 1.0 |
| Reynolds number reference length/m | $L$ | 2.0 |
| Reynolds numbers/- | $Re$ | $15.216 \times 10^{6}$ [†] │ $30.432 \times 10^{6}$ [‡] |
| Relative Cauchy error for … | | |
| …lift coefficient/- | $\epsilon_{C_L}$ | $1 \times 10^{-6}$ |
| …drag coefficient/- | $\epsilon_{C_d}$ | $1 \times 10^{-6}$ |
| …lateral force/- | $\epsilon_{Fy}$ | $1 \times 10^{-4}$ |
| …longitudinal moment coefficient/- | $\epsilon_{C_{mx}}$ | $1 \times 10^{-3}$ |
| …lateral moment coefficient/- | $\epsilon_{C_{my}}$ | $5 \times 10^{-6}$ |

[†] Mach 0.4; [‡] Mach 0.8.

The procedure for the presented studies is illustrated in Figure 2. The structural mode shapes usually involved in T-tail flutter, i.e., VTP out-of-plane bending and torsion, and their quadratic deformation components are approximated by rigid body rotations with respect to the longitudinal axis for the VTP out-of-plane bending and the vertical axis for the VTP torsion. The origin for the rotational deformations is at the VTP root. This allows for a straightforward, analytical evaluation of the linear and quadratic deformation components from rotation matrices without the need for using a structural solver to compute the higher-order deformation components. With this, errors in computing the deformation components are avoided and the terms involved in the deformation process are explicitly defined. Sections 2.1 and 2.2 will outline details regarding the extended modal approach and the method chosen to obtain quadratic displacement components. The HTP is subjected to harmonic forced motion within a CFD framework in the time domain at five displacement amplitudes and two reduced frequencies. Inviscid flow computations using Euler equations were carried out at a Mach number of 0.4, while viscous flow computations using RANS equations with the negative Spalart–Allmaras turbulence model [24] cover the Mach numbers of 0.4 and 0.8; see Section 2.3 for further details. During the runtime of the CFD solution, the unsteady aerodynamic forces are generalized employing the linear, as well as the extended modal formulation. The resulting time domain data are assessed regarding their frequency content and evaluated in terms of aerodynamic stiffness and damping; see Section 2.4.

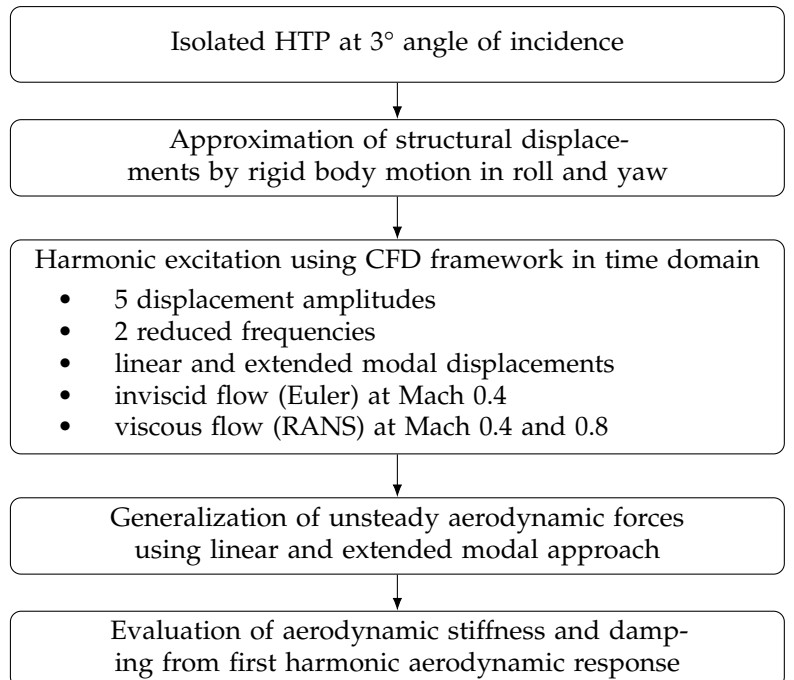

**Figure 2.** Approach.

### 2.1. Extended Modal Approach

The geometrically nonlinear physical displacements of a structural grid point are formulated in terms of the linear mode shapes $\boldsymbol{\phi}_i$ and the corresponding quadratic mode shape components $\boldsymbol{g}_{ij}$ [19], viz.

$$x = \sum_{i=1}^{m} \boldsymbol{\phi}_i q_i + \sum_{i=1}^{m} \sum_{j=1}^{m} \boldsymbol{g}_{ij} q_i q_j \tag{1}$$

where $q_{i,j}$ are the $i$th and the $j$th modal degree of freedom, respectively. The aerodynamic forces are generalized with respect to the modal degree of freedom $p$ according to

$$^{p}Q = {}^{p}\boldsymbol{\phi}^{T} f + {}^{p}\boldsymbol{g}^{i^{T}} f q_i \tag{2}$$

with summation over repeated index $i$ and

$$^{p}Q^{(0)} = {}^{p}\boldsymbol{\phi}^{T} f \tag{3}$$

as the linear GAF term. The extended modal equation of motion for degree of freedom $p$ becomes

$$^{p}M^{i} \ddot{q}_i + \left( {}^{p}K^{i} - {}^{p}\boldsymbol{g}^{i^{T}} f \right) q_i = {}^{p}\boldsymbol{\phi}^{T} f \tag{4}$$

The quadratic mode shape components are distinguished into uncoupled and coupled quadratic mode shape components. Uncoupled quadratic mode shape components result only from a single linear mode shape and add to the diagonal elements of the generalized stiffness matrix. Coupled quadratic mode shape components are subject to two linear mode shapes and introduce a mode coupling by means of off-diagonal generalized stiffness matrix elements depending on the force vector $f$. In the context of this work, however, only uncoupled quadratic mode shape components will be considered.

### 2.2. Obtaining Quadratic Mode Shape Components

While the linear mode shapes $\boldsymbol{\phi}_i$ in Equation (1) may be obtained from conventional solutions to an eigenvalue problem characterized by the mass and stiffness of the structure,

the quadratic mode shape components $g_{ij}$ require alternative approaches. Besides using linear or nonlinear finite element analysis [18,19,25], an approximation of the linear mode shapes by rigid body rotations may already be suitable to obtain reasonable linear mode shapes, as well as their quadratic displacement components. In addition, the modal approach commonly used in numerical flutter assessment can be maintained. Focusing exemplarily on the roll motion, the analytical description of the displacement uses the nonlinear rotation matrix with roll angle $\varphi$:

$$\boldsymbol{R} = \begin{bmatrix} 1 & 0 & 0 \\ 0 & cos(\varphi) & -sin(\varphi) \\ 0 & sin(\varphi) & cos(\varphi) \end{bmatrix} \tag{5}$$

Expanding the sine and cosine terms in Equation (5) as Taylor series with collected Higher-Order Terms (H.O.T) [26]:

$$sin(\varphi) \approx \varphi - \frac{\varphi^3}{3!} + \text{H.O.T} \tag{6}$$

$$cos(\varphi) \approx 1 - \frac{\varphi^2}{2!} + \text{H.O.T} \tag{7}$$

and truncating them after the first- and second-order terms, respectively, lead to the linear (Equation (8), superscript $()^{(1)}$) and the quadratic (Equation (9), superscript $()^{(2)}$) rotation matrices.

$$\boldsymbol{R}^{(1)} = \begin{bmatrix} 1 & 0 & 0 \\ 0 & 1 & -\varphi \\ 0 & \varphi & 1 \end{bmatrix} \tag{8}$$

$$\boldsymbol{R}^{(2)} = \begin{bmatrix} 1 & 0 & 0 \\ 0 & 1 - 1/2\varphi^2 & -\varphi \\ 0 & \varphi & 1 - 1/2\varphi^2 \end{bmatrix} \tag{9}$$

The linear mode shape and its quadratic displacement component become

$$\boldsymbol{\phi} = \frac{\partial \boldsymbol{R}^{(1)}}{\partial \varphi} \boldsymbol{x} \tag{10}$$

and

$$\boldsymbol{g}_{\varphi\varphi} = \frac{1}{2} \frac{\partial^2 \boldsymbol{R}^{(2)}}{\partial \varphi^2} \boldsymbol{x} \tag{11}$$

with

$$\frac{\partial \boldsymbol{R}^{(1)}}{\partial \varphi} \boldsymbol{x}_i = \begin{bmatrix} 0 & 0 & 0 \\ 0 & 0 & -1 \\ 0 & 1 & 0 \end{bmatrix} \boldsymbol{x}_i \tag{12}$$

$$\frac{\partial^2 \boldsymbol{R}^{(2)}}{\partial \varphi^2} \boldsymbol{x}_i = \begin{bmatrix} 0 & 0 & 0 \\ 0 & -1 & 0 \\ 0 & 0 & -1 \end{bmatrix} \boldsymbol{x}_i \tag{13}$$

and $\boldsymbol{x}$ as the vector of surface grid point coordinates. Note that, when using orthogonal rotation vectors, the coupled quadratic mode shape components are zero.

The linear and quadratic displacement components are visualized in Figure 3 for the HTP roll and yaw motion. The blue surface color illustrates the linear displacements and the orange surface color the quadratic displacement components against the undisplaced geometry shown in gray. Adding these components results in a second-order approximation of the nonlinear displacement field. For the roll motion (Figure 3a), the quadratic displacement components result in a vertical motion of the HTP in combination with a re-

duction in span. The second-order displacement components of the yaw motion (Figure 3b) is a reduction in the span and chord length of the HTP.

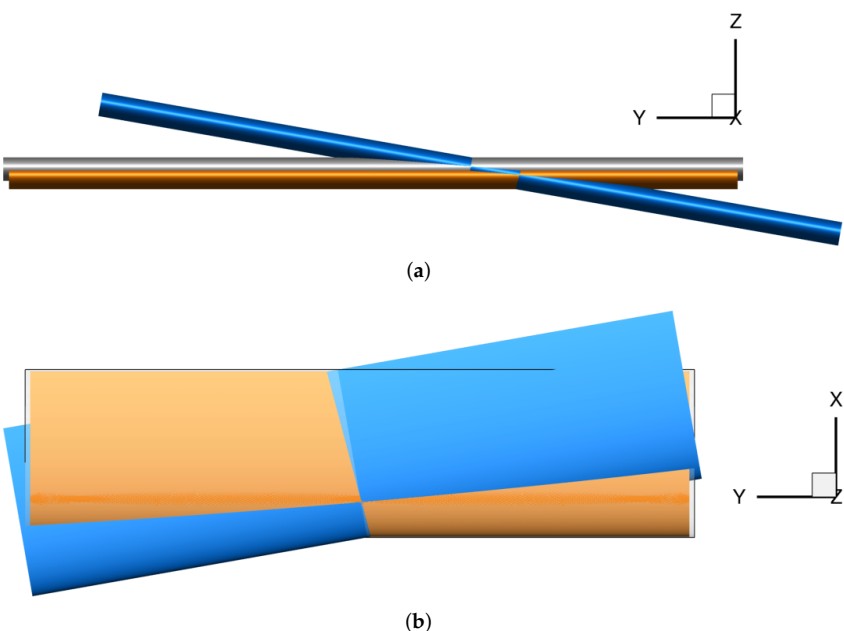

(a)

(b)

**Figure 3.** Linear rigid body displacements (blue) and quadratic displacement components (orange) against undisplaced geometry (gray). (**a**) Roll motion; (**b**) yaw motion.

### 2.3. Time Domain Harmonic Forced Motion

Harmonic excitation of the CFD surface mesh is used for computing the aerodynamic response at varying amplitudes and frequencies; see Table 1. A Cauchy convergence criterion with relative errors $\epsilon$ is used for the inner CFD iterations at each physical time step on lift coefficient $C_L$, drag coefficient $C_d$, and side force $F_y$, as well as on the coefficients for the moment around the longitudinal axis $C_{m_x}$ and lateral axis $C_{m_y}$. The simulations cover the low reduced frequency of 0.056 together with the high reduced frequency of 0.231, as well as the roll and, in the case of the inviscid flow studies, also the yaw angle amplitudes of 0.010°, 0.917°, 1.834°, 3.669°, and 5.000°. Taking fully nonlinear displacements as a basis, the roll angle amplitudes correspond to relative displacements of the center of the HTP with respect to the VTP span of 0.017%, 1.600%, 3.200%, 6.403%, and 8.724%, respectively. The largest displacement amounts to roughly 0.6 m at the HTP tip.

The resulting time domain aerodynamic forces are generalized according to the linear modal formulation (3) and the extended modal formulation (2). A discrete Fourier transform (DFT) algorithm is applied to a sliding time window with a size of two periods of oscillation to continue the simulation until the magnitudes of the target Generalized Aerodynamic Forces (GAFs) show a convergence with a residual of 0.1% for a time span of two periods. Figure 4 illustrates this approach for the hysteretic generalized aerodynamic response to harmonic forced motion shown in the upper left plot. The development of the GAF magnitude with the sliding DFT window is shown in the lower left plot and indicates a quick convergence. The right-hand-side plots depict the GAF magnitude and phase angle extracted from the last two periods of oscillation. Only a first harmonic GAF content is evident, which is used to further analyze aerodynamic stiffness and damping.

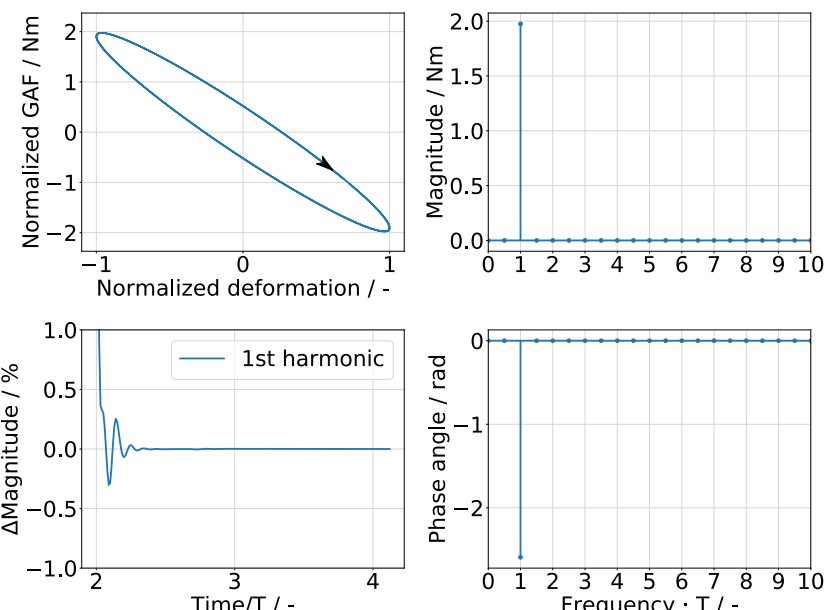

**Figure 4.** Analysis approach of generalized aerodynamic response.

### 2.4. Aerodynamic Stiffness and Damping from Time Domain Results

Recording the generalized aerodynamic forces in response to the harmonic motion input of a generalized coordinate results in a hysteretic aerodynamic response; see Figure 5. Here, the fictional input signal $q(t)$ is plotted against time $t$ in the bottom figure, and the upper left figure shows the time history of the fictional output signal $Q(t)$. The top right figure displays the resulting hysteretic response. These hystereses are analyzed regarding aerodynamic stiffness and damping of the first harmonic contents based on their magnitudes and phase angles. In general, the mechanical work of a system with hysteretic response to harmonic excitation may be considered as consisting of a contribution due to the system's stiffness term ($W_k$) and one due to the system's damping term ($W_c$) [27,28]. The inclination and enclosed area of the hysteresis, as shown in Figure 5, are the parameters defining these work terms. By focusing on the first harmonic content in the signals and evaluating the corresponding integrals, the normalized work terms become [29]

$$\frac{W_k}{\hat{q}^2} = \frac{1}{\hat{q}^2} \int_0^{\hat{q}} \frac{\hat{Q}\cos(\varphi)}{\hat{q}} q \mathrm{d}q = \frac{1}{2} \frac{\hat{Q}}{\hat{q}} \cos(\delta) \tag{14}$$

$$\frac{W_c}{\hat{q}^2} = \frac{1}{\hat{q}^2} \int_T^{T+\frac{2\pi}{\omega}} Q \frac{\mathrm{d}q}{\mathrm{d}t} \mathrm{d}t = \frac{\hat{Q}}{\hat{q}} \pi \sin(\delta) \tag{15}$$

with $\delta$ being the phase difference between the output and input signal, $T$ the period of oscillation, $\omega$ the angular frequency, and $\hat{q}$ and $\hat{Q}$ the magnitudes of input and output signal, respectively. In structural dynamics, the concept of a complex stiffness is commonly employed to describe the stiffness and damping characteristics with a single complex-valued quantity, usually referred to as a complex modulus. It consists of a real part, the mechanical storage stiffness, and an imaginary part, the mechanical loss stiffness. The complex modulus is defined as

$$k^*(\omega) = k'(\omega) + \mathrm{j}k''(\omega) \tag{16}$$

$$k'(\omega) = \frac{\hat{Q}}{\hat{q}} \cos(\delta) = \frac{2W_k}{\hat{q}^2} \tag{17}$$

$$k''(\omega) = \frac{\hat{Q}}{\hat{q}} \sin(\delta) = \frac{W_c}{\pi\hat{q}^2} \tag{18}$$

$k'(\omega)$ characterizes the stiffness property and $k''(\omega)$ the damping property. As both quantities are merely a scaling of the integrals described in Equations (14) and (15), this concept is used in the present work to assess the aerodynamic stiffness and damping from harmonic forced excitations.

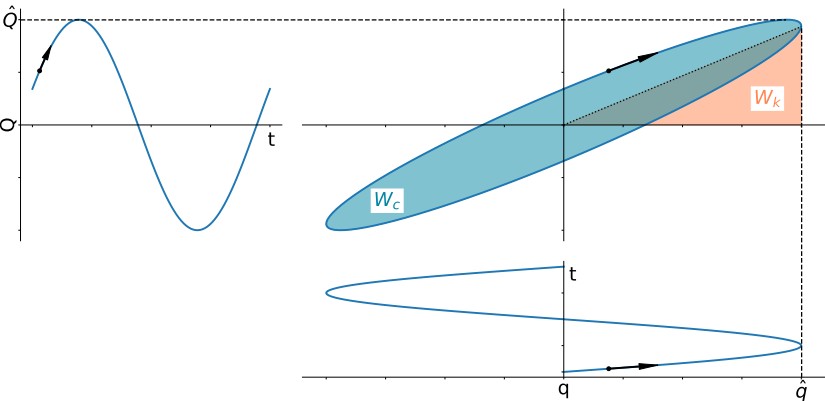

**Figure 5.** Illustration for GAF hysteresis analysis.

### 2.5. CFD Models

The study targeting aerodynamic nonlinearities due to HTP roll and yaw motion comprises inviscid, as well as viscous flow computations. Hence, two CFD mesh topologies are required. The CFD mesh used for inviscid flow computations is described first, followed by that used for viscous flow computations. All CFD meshes have in common an initially semi-span mesh, which is mirrored to facilitate a symmetric CFD mesh and to avoid numerical asymmetries. The farfield covers 50 chord lengths in the front, left, right, below, and above the configuration, as well as 150 chord lengths aft of it.

#### 2.5.1. Inviscid Flow

Illustrated in Figure 6 is the surface mesh of the left semi-span geometry.

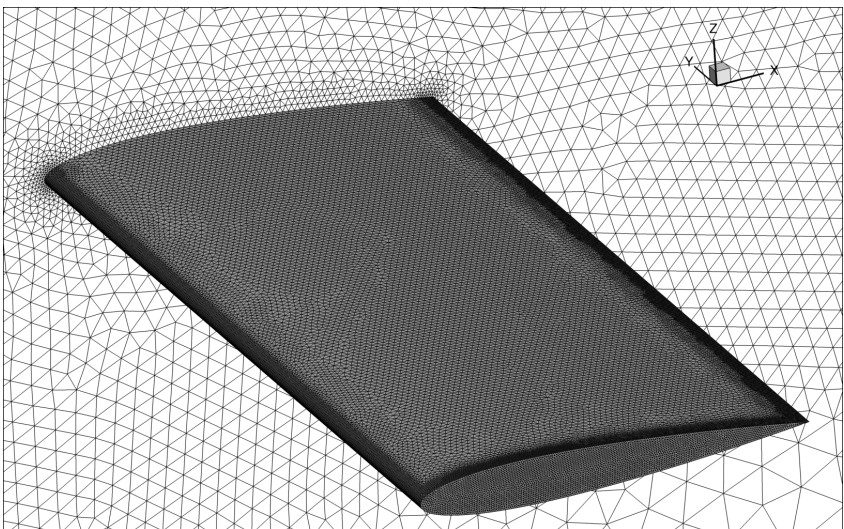

**Figure 6.** CFD mesh of the isolated HTP for inviscid flow computations.

The mesh consists of roughly 865 k nodes and 4.8 million tetrahedral elements and is a result of a mesh independency study focusing on the complex GAF values at the largest displacement input and high reduced frequency. Three meshes with the grid point and element counts listed in Table 2 form the basis for the independency study. The resulting deviations $\Delta$ in the magnitudes and phase angles of the first harmonic GAF contents with respect to the fine mesh are illustrated in Figure 7. Here, $Q_{hh}(1,1)$ denotes the aerodynamic

forces generalized with the roll motion and $Q_{hh}(2,1)$ the aerodynamic forces generalized with the yaw motion. The comparison substantiates the medium mesh density to be sufficiently accurate.

**Table 2.** Mesh densities used for mesh independency study.

| Parameter | Coarse | Medium | Fine |
|---|---|---|---|
| Number of grid points | 0.432 million | 0.865 million | 1.412 million |
| Number of surface triangles | 85.732 k | 178.066 k | 412.094 k |
| Number of volume tetrahedrons | 2.407 million | 4.816 million | 7.689 million |

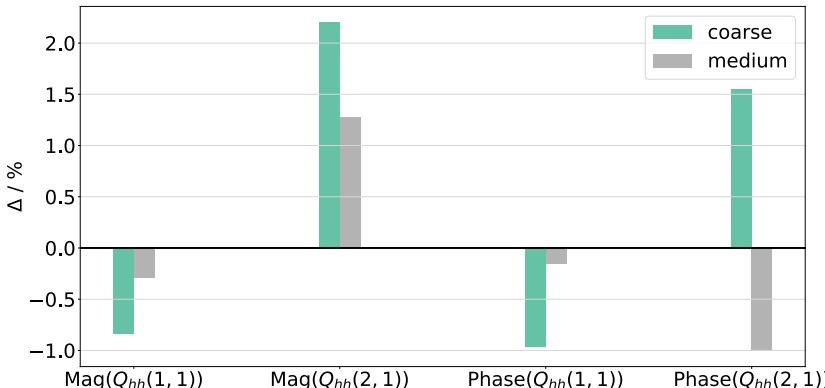

**Figure 7.** Deviations in magnitudes and phase angles of first harmonic GAF contents with respect to fine mesh (inviscid flow, Mach 0.4, reduced frequency 0.231, 5.0° roll angle amplitude).

2.5.2. Viscous Flow

The computational mesh for viscous flow simulations differs from that used for inviscid flow computations by a structured surface mesh and an additional discretization of the boundary layer. The latter requires knowledge of the turbulent boundary layer thickness at the trailing edge, which is calculated after [30]. With the reference values as listed in Table 1 and

$$Re = \frac{Ma\sqrt{\kappa RT}L}{\nu} \tag{19}$$

$$\frac{\delta(L)}{L} = \frac{0.37}{Re^{0.2}} \tag{20}$$

the Reynolds number $Re$ ranges from roughly $15 \times 10^6$ to $30 \times 10^6$ and the boundary layer thickness $\delta$ at the trailing edge amounts to roughly 0.027 m and 0.024 m, respectively.

Estimating the first layer thickness with a desired $y^+$-value of 1 according to [31] with

$$\Delta y_1 = \frac{y^+ \nu}{v_\tau} \tag{21}$$

$$v_\tau = \sqrt{\frac{\tau_w}{\rho}} \tag{22}$$

$$\tau_w = 0.5 C_f \rho v^2 \tag{23}$$

$$C_f \approx 0.058 Re^{-0.2} \tag{24}$$

$$v = Ma\sqrt{\kappa RT} \tag{25}$$

results in a minimum first layer thickness of $4.034 \times 10^{-6}$ m for a Mach number of 0.4 and $2.162 \times 10^{-6}$ m for a Mach number of 0.8. After manual iterations, a value of $1.4 \times 10^{-6}$ m is used for all Mach numbers to ensure a proper resolution of the boundary layer. The final parameters for the prism layer are listed in Table 3.

**Table 3.** Parameters of prism layer.

| Parameter | Value |
| --- | --- |
| First layer thickness | $1.4 \times 10^{-6}$ m |
| Number of layers | 38 |
| Stretching ratio | 1.25 |
| Total prism layer height | 0.027 m |

With the bounding box as described above, the finally used mesh, shown in Figure 8, consists of roughly 2.2 million nodes and 5.7 million volume elements.

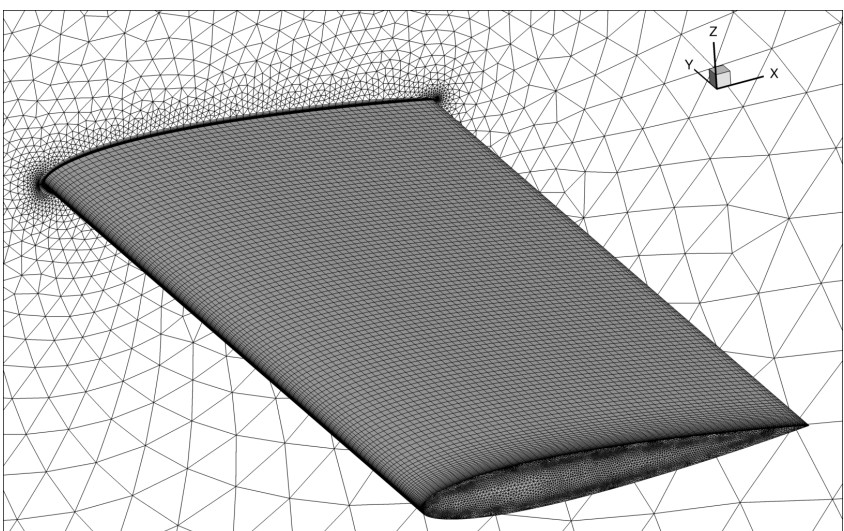

**Figure 8.** CFD mesh of the isolated HTP for viscous flow computations.

*2.6. Temporal Discretization*

Using harmonic forced motion of the CFD surface mesh with a succeeding evaluation of the converged frequencies contained in the generalized aerodynamic forces signal requires a proper time step size along with a reasonable simulation duration to ensure that the transients have faded away. The latter is handled by the sliding DFT algorithm combined with the GAF convergence criterion, as illustrated in Figure 4. The time step size is relevant as it prescribes the gradient of the displacement, which in turn impacts the convergence of the inner iterations required by the CFD solver. As the convergence is handled by the Cauchy criteria defined for the aerodynamic coefficients of interest, the time step size does not alter the final results, but the computational time. Hence, apart from the largest displacements listed in Table 1, a time step size of $2 \times 10^{-3}$ s was found to be adequate. The simulations with large displacement amplitude and inviscid flow conditions require an adaption of the time step size to $5 \times 10^{-4}$ s to maintain a reasonable computational time.

## 3. Results

At first, the results for inviscid flow at a Mach number of 0.4 are presented in terms of GAF hystereses and aerodynamic stiffness and damping obtained from evaluating the first harmonic aerodynamic response to harmonic forced motion. The impact of fluid viscosity on the aerodynamic response is addressed next, followed by its sensitivity to compressibility with consideration of a high Mach number of 0.8.

*3.1. Inviscid Flow*

3.1.1. GAF Hystereses

The GAF response over sinusoidal motion input in roll and yaw for increasing amplitudes and reduced frequency values $k$ is summarized in Figure 9. Here, Figure 9a,c

illustrate the aerodynamic influence of the HTP roll motion on itself and on the yaw motion, $Q_{hh}(1,1)$ and $Q_{hh}(2,1)$. The aerodynamic influence of the HTP yaw motion on the roll motion and on itself, $Q_{hh}(1,2)$ and $Q_{hh}(2,2)$, is depicted in Figure 9b,d, respectively. For all figures, the reduced frequency is increased from top to bottom, whereas the results based on linear and quadratic displacement are displayed from left to right. The input signal and the response are normalized to the displacement amplitude. Figure 9a shows largely coinciding ellipsoids with regard to the displacement amplitude for the diagonal GAF term $Q_{hh}(1,1)$. A distinct impact of the nonlinear displacement term on the inclination of the hysteresis is notable, even for the smallest amplitudes. At a high reduced frequency, this change in inclination is less pronounced compared to that at a low reduced frequency, but still observable. As outlined in Section 2.4, the inclination is a measure for the aerodynamic stiffness, and hence, a reduction in aerodynamic stiffness by the addition of second-order displacement terms is shown. This non-zero stiffness resulting from fully described unsteady aerodynamic forces in combination with a linear displacement model has already been observed by L. van Zyl in [16] and is termed "spurious stiffening". As this effect is purely numerical, the quadratic displacement terms are required to properly describe the physical system even for linear T-tail flutter assessment. Apart from the identical change in stiffness for all deformation amplitudes, the elliptical shapes of the hystereses do not change considerably with amplitude and, hence, show a linear harmonic input–output behavior. Contrary to the observations made for $Q_{hh}(1,1)$, the shapes of the hystereses of $Q_{hh}(2,1)$ shown in Figure 9c are majorly affected by higher-order displacement terms. For linear displacements, a distinct higher-order term in the aerodynamic response is notable at large displacement amplitudes, which is not present at small displacement amplitudes. This is detailed for one period of oscillation in Figure 10a for the smallest displacement amplitude and Figure 10b for the largest one. This higher-order term in the aerodynamic response is reduced with quadratic displacement components. However, a deviation of the hystereses from an elliptical shape with increasing displacement amplitude can still be identified for quadratic displacements at high reduced frequencies, indicating as well a higher harmonic content in the aerodynamic response. Insensitive to the displacement description is the aerodynamic coupling term $Q_{hh}(1,2)$; see Figure 9b. For both linear and quadratic displacements, a small reduction in inclination with increasing displacement amplitude can be noticed. For all reduced frequencies and amplitudes, the hystereses maintain their elliptical shape. Similar to the first diagonal the GAF term, the second diagonal term $Q_{hh}(2,2)$ (Figure 9d) shows a change in inclination for all amplitudes and reduced frequencies when higher-order displacement terms are taken into account. However, the VTP, which is not modeled in this study, will presumably induce aerodynamic forces that are not negligible for this GAF term. Thus, the results presented for this particular GAF matrix element must be assessed cautiously. Besides this, all hystereses show an elliptical shape and coincide for all displacement amplitudes, demonstrating a linear harmonic input–output behavior.

### 3.1.2. Aerodynamic Stiffness and Damping

With the approach outlined in Section 2.4, the aerodynamic stiffness and damping characteristics are evaluated in terms of relative deviations to the values for linear displacements at the smallest displacement amplitude, as this represents the values used for linear flutter assessment. Since all GAF terms except for that shown in Figure 9c appear to be rather insensitive to the displacement amplitude, the analysis will be focused on the off-diagonal GAF term $Q_{hh}(2,1)$, i.e., the mechanical work performed on the HTP yaw motion by aerodynamic forces induced by HTP roll motion. Figure 11 shows the deviations in stiffness ($\Delta k'$) and damping ($\Delta k''$) over the displacement amplitude evaluated for the first harmonic term in the GAF signal. A deviation of zero indicates that the results agree with those at the smallest displacement amplitude, and hence, a linear response is shown. The results based on the linear modal approach are represented by the blue solid line, while the orange solid line illustrates the results based on the extended modal approach.

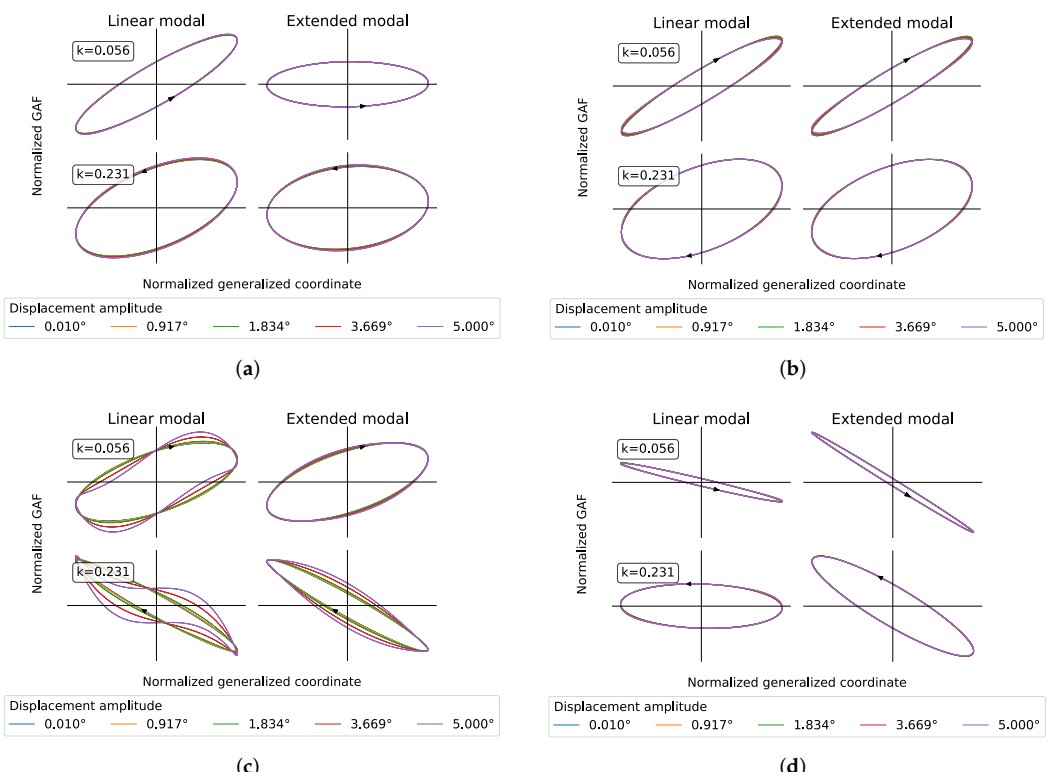

**Figure 9.** GAF hystereses (inviscid flow, Mach 0.4). (**a**) $Q_{hh}(1,1)$; (**b**) $Q_{hh}(1,2)$; (**c**) $Q_{hh}(2,1)$; (**d**) $Q_{hh}(2,2)$.

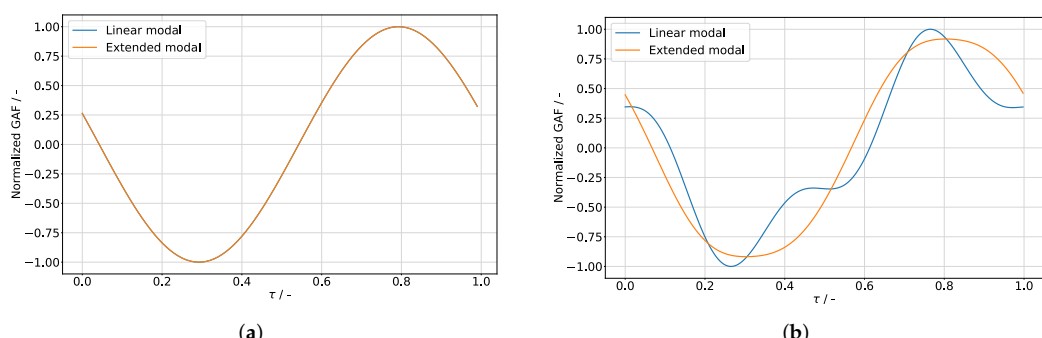

**Figure 10.** Time history of $Q_{hh}(2,1)$ for one period of oscillation (inviscid flow, Mach 0.4, reduced frequency 0.231): (**a**) 0.01° roll motion amplitude; (**b**) 5.0° roll motion amplitude.

Identical stiffness and damping values are shown for all reduced frequencies at the smallest displacement amplitudes. As elaborated, the simulations covered only uncoupled quadratic mode shape components, as the coupled quadratic mode shape components are zero. At a small displacement amplitude, hence, the stiffness is not affected by second-order displacement terms. Increasing the displacement amplitude reveals a nonlinear dependency of the aerodynamic stiffness (left figure column) with a sign change between linear and extended modal displacements. While an increase in stiffness for low reduced frequencies is observable for the linear modal displacement, the extended modal displacement actually indicates a decrease in stiffness, which is significantly lower in magnitude compared to the linear approach. At high reduced frequencies, the linear displacement results showed a reduction in stiffness up to 15% at a 5.0° displacement amplitude. With quadratic displacement components, an increase in stiffness is observable, but again considerably lower in magnitude compared to the change in stiffness based on linear displacements. The extended modal approach linearizes the aerodynamic stiffness at both reduced frequencies.

The impact of the higher-order displacement components on aerodynamic damping (right figure column) is again a sign change for low reduced frequencies. The linear modal approach results in a slight reduction in damping, whereas the quadratic displacement terms yield an increase in damping with a higher magnitude. At high reduced frequencies, both displacement approaches show a distinctly nonlinear dependency of the damping on the displacement amplitude with deviations exceeding 40% for the largest displacement amplitude. Here, the quadratic displacement approach results in a larger impact on the damping compared to the linear one.

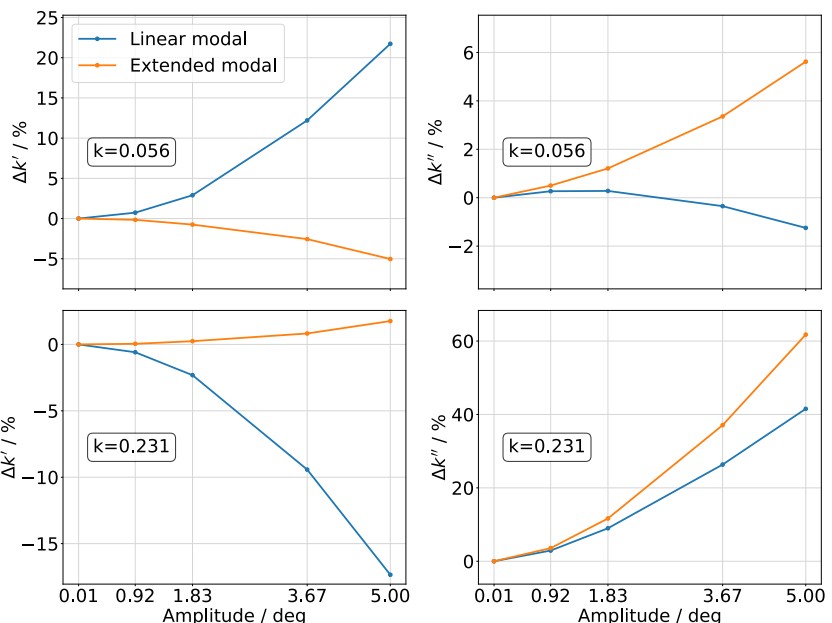

**Figure 11.** Aerodynamic stiffness and damping over amplitude for $Q_{hh}(2, 1)$ (inviscid flow, Mach 0.4).

The essential observations regarding the stiffness and damping of the aerodynamic coupling term at inviscid subsonic flow and the impact of geometric nonlinearities on these values may be summarized as follows:

- Aerodynamic stiffness:
  - Nonlinear for low and high reduced frequencies with linear displacements;
  - Only marginally nonlinear for low and high reduced frequencies with quadratic displacements.
- Aerodynamic damping:
  - Marginally nonlinear for low reduced frequencies and both displacement descriptions;
  - Nonlinear for high reduced frequencies and both displacement descriptions;
  - Increased nonlinearity with quadratic displacements.

*3.2. Impact of Fluid Viscosity*

The viscous flow computations concentrate on the aerodynamic response to HTP roll motion only. At first, a low Mach number case is considered to support a comparison to the inviscid flow results. Section 3.3 then outlines the results for viscous transonic flow.

### 3.2.1. GAF Hystereses

The impact of fluid viscosity on the aerodynamic response to HTP roll motion is depicted in Figure 12. On the left-hand side, Figure 12a,c repeat the results for inviscid flow as presented in Section 3.1.1, while the right-hand side contrasts the results for viscous flow in Figure 12b,d. As for the inviscid flow results, a reduction in aerodynamic stiffness is observable for $Q_{hh}(1, 1)$ at both reduced frequencies when quadratic displacement components are considered. Regarding this GAF matrix element, the results agree well

between inviscid and viscous flow for all displacement amplitudes and both reduced frequencies. On the contrary, the aerodynamic coupling term $Q_{hh}(2,1)$ appears to be affected by fluid viscosity. At low reduced frequency $k = 0.056$, the inclinations and areas of the hystereses change noticeably, indicating an impact on both aerodynamic stiffness and damping. At high reduced frequency $k = 0.231$, the impact is distinctly lower, especially regarding the numerical results based on the extended modal approach. Nevertheless, the aerodynamic response is still revealed to be nonlinear with regard to the displacement amplitude and, as observed for the inviscid flow, shows a higher harmonic content. This content is dominant for the results based on linear displacements, but less pronounced for the results based on nonlinear displacements.

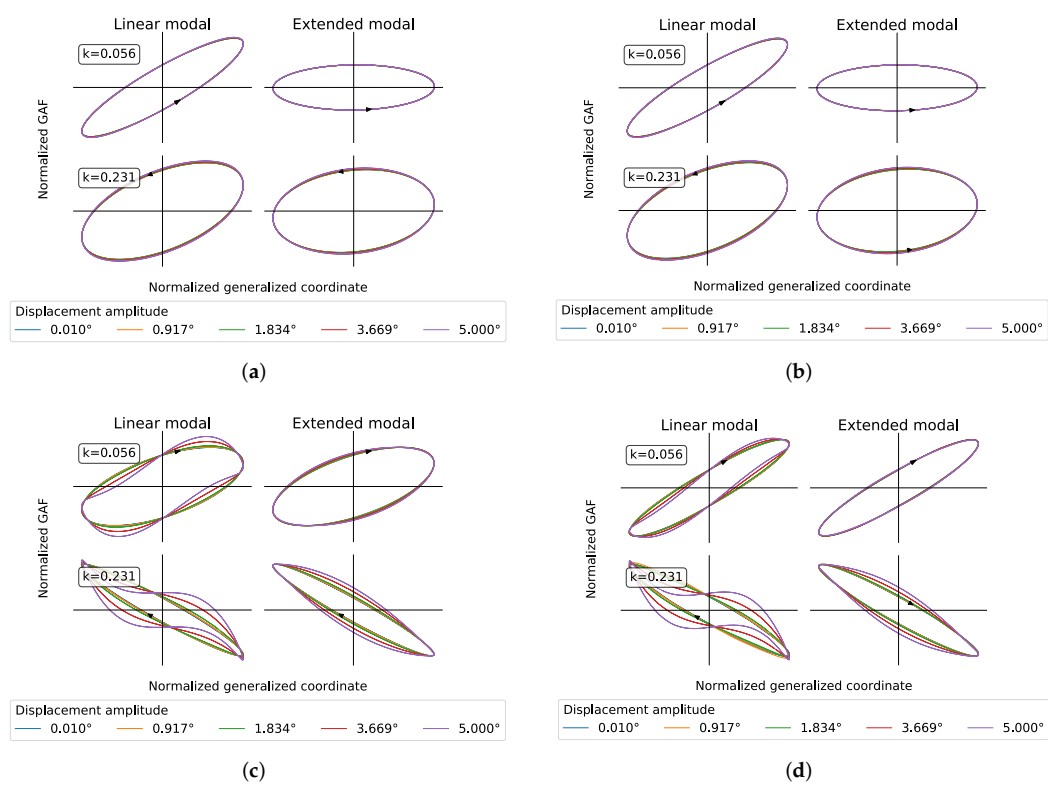

**Figure 12.** Impact of fluid viscosity on GAF hystereses (Mach 0.4). (**a**) Inviscid flow—$Q_{hh}(1,1)$; (**b**) viscous flow—$Q_{hh}(1,1)$; (**c**) inviscid flow—$Q_{hh}(2,1)$; (**d**) viscous flow—$Q_{hh}(2,1)$.

### 3.2.2. Aerodynamic Stiffness and Damping

Following the observations made above, the off-diagonal GAF term $Q_{hh}(2,1)$ is again selected for further studies regarding the impact of fluid viscosity on aerodynamic stiffness and damping, which is depicted in Figure 13. The aerodynamic stiffness, illustrated on the left-hand side, reveals a distinct increase in value for low reduced frequencies even at the lowest displacement amplitudes. At high reduced frequencies, a less pronounced reduction in value can be observed. Albeit the stiffness reduction identified for the high reduced frequency solution is by roughly one order of magnitude lower in absolute value compared to the low reduced frequency case, it still amounts to roughly a 13% deviation from the inviscid flow results. However, the general trend of the stiffness values with increasing displacement amplitude agrees between the inviscid and viscous flow results and both displacement descriptions. That is, the impact of fluid viscosity on the aerodynamic stiffness is independent of the displacement amplitude and the geometric nonlinearity introduced by the extended modal approach. Regarding the aerodynamic damping shown on the right-hand side of Figure 13, the impact of fluid viscosity is amplitude dependent, but again, the general effects of an increasing displacement amplitude agree with the inviscid flow results. For the low reduced frequency results at small displacement amplitudes, the

viscous aerodynamic forces show a marginal increase of 2% in damping compared to the inviscid results. Increasing the displacement amplitude reveals that the viscous forces yield a stronger drop in aerodynamic damping compared to the linear modal displacement results. With the addition of geometrically nonlinear displacements, the viscous forces reduce the nonlinear character of the aerodynamic damping. At the high reduced frequency, the general trends of the aerodynamic damping terms agree well between the inviscid and viscous flow results. However, the viscous terms yield a weakened nonlinear characteristic of the aerodynamic damping.

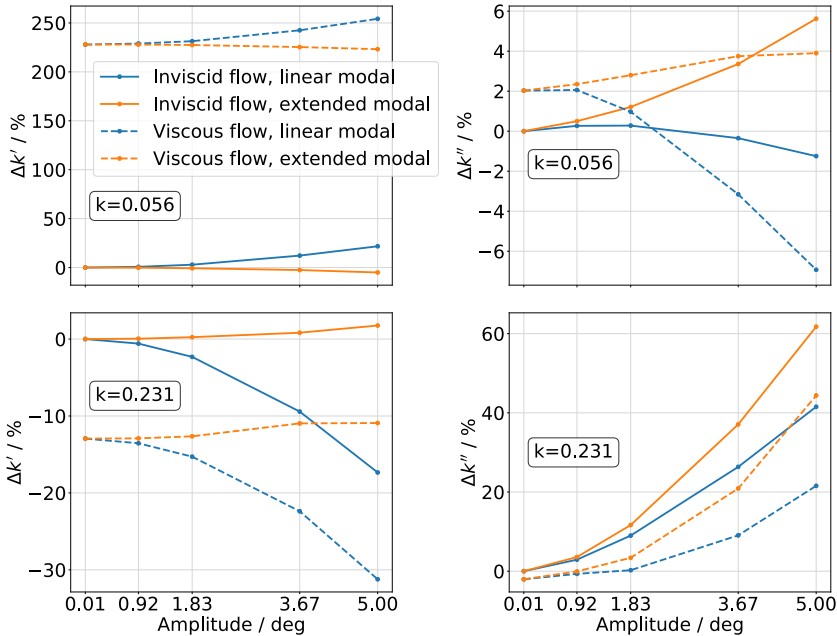

**Figure 13.** Impact of fluid viscosity on aerodynamic stiffness and damping of $Q_{hh}(2,1)$ (Mach 0.4).

The essential observations regarding the impact of fluid viscosity on stiffness and the damping of the aerodynamic coupling term at a moderate subsonic Mach number may be summarized as follows:

- Aerodynamic stiffness:
    - Significant and amplitude-independent offset with respect to inviscid flow results;
    - No remarkable impact on nonlinear character.
- Aerodynamic damping:
    - Marginally larger damping at small displacement amplitudes and low reduced frequencies;
    - Marginally lower damping at small displacement amplitudes and high reduced frequencies;
    - Increased nonlinearity for linear modal displacement at low reduced frequencies;
    - Decreased nonlinearity for extended modal displacement at low reduced frequencies;
    - Decreased nonlinearity for linear and extended modal displacement at high reduced frequencies.

### 3.3. Impact of Fluid Compressibility

Having addressed the effect of fluid viscosity on the aerodynamic coupling term, an increase in Mach number from 0.4 to 0.8, while maintaining viscous flow conditions shall give insight into the sensitivity of the nonlinearity to fluid compressibility. Considering exemplarily the maximum allowable Mach number of a Gulfstream G650 of 0.925, the Mach number of 0.8 selected for the simulations corresponds closely to the normal Mach number of the 30° swept HTP [32]. Hence, the flow conditions come close to real-world applications.

### 3.3.1. GAF Hystereses

Figure 14 presents the results for a Mach number of 0.8 on the right-hand side in comparison to the already presented results for a Mach number of 0.4 on the left-hand side. Consistent with the observations made above for the inviscid, as well as viscous flow results at Mach number 0.4, the diagonal GAF term $Q_{hh}(1,1)$ shows the elaborated impact of geometric nonlinearities on aerodynamic stiffness. Additionally, a visually identical aerodynamic response between subsonic and transonic speeds is evident, which indicates that this GAF matrix element is insensitive to fluid viscosity and compressibility. Only the aerodynamic coupling term $Q_{hh}(2,1)$ shows strong sensitivities to the parameter variation. Here, especially, the higher harmonic content is affected and significantly reduced at high Mach number. Furthermore, a distinct increase in the area of the hystereses is notable, which indicates an increase in aerodynamic damping. A change in inclination is observable as well. Contrary to the results presented so far, however, the geometric nonlinearity is evidently only of minor relevance for this GAF matrix element at transonic speed.

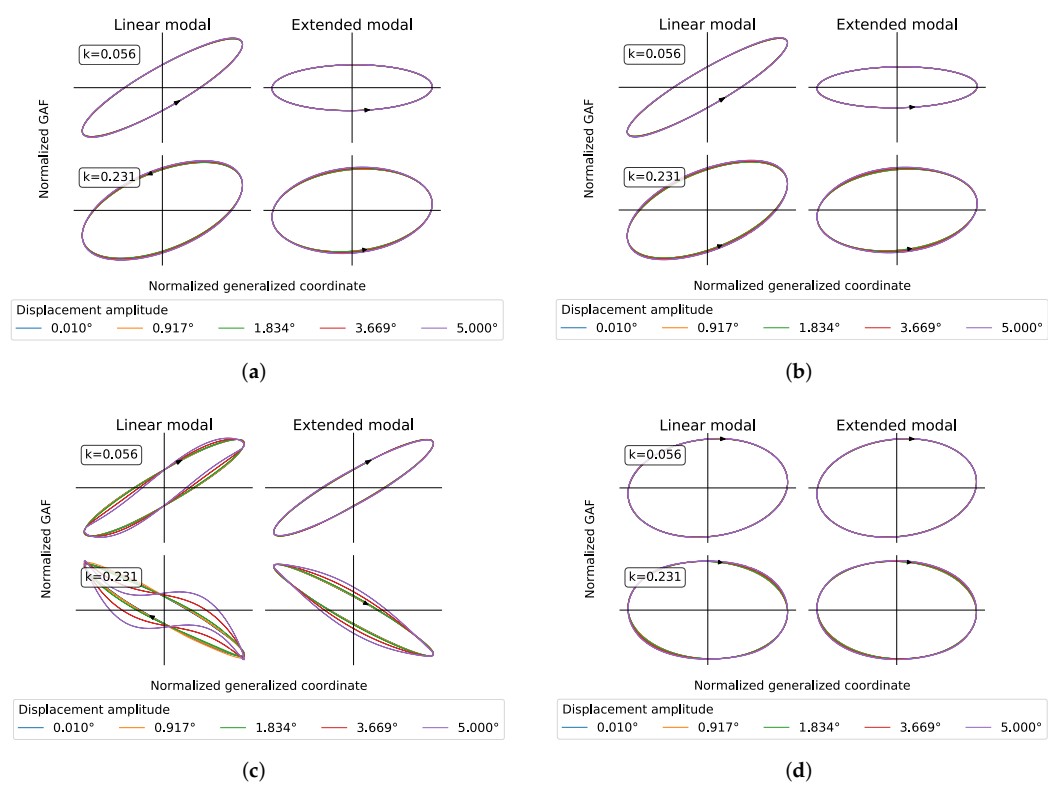

**Figure 14.** Impact of fluid compressibility on GAF hystereses (viscous flow). (**a**) Mach 0.4—$Q_{hh}(1,1)$; (**b**) Mach 0.8—$Q_{hh}(1,1)$; (**c**) Mach 0.4—$Q_{hh}(2,1)$; (**d**) Mach 0.8—$Q_{hh}(2,1)$.

### 3.3.2. Aerodynamic Stiffness and Damping

Focusing again on the aerodynamic coupling term with the hystereses shown in Figure 14d and addressing aerodynamic stiffness and damping require considering the deviations from the smallest displacement results individually for Mach 0.4 and Mach 0.8, respectively, as the aerodynamic response is, in general, nonlinear with Mach number. Figure 15 illustrates the impact of fluid compressibility on the nonlinear response. In general, the aerodynamic stiffness nonlinearity is amplified at high Mach number, whereas the damping nonlinearity is almost entirely canceled. Additionally, the impact of the higher-order displacement components on the nonlinearities is distinctly reduced. While the low Mach number results presented in Sections 3.1 and 3.2 suggested an almost complete cancellation of the stiffness nonlinearity with the addition of geometric nonlinearity, the high Mach number case leaves this observation valid only at low reduced frequency.

However, a mild mitigation of the stiffness nonlinearity due to nonlinear displacement components is still apparent.

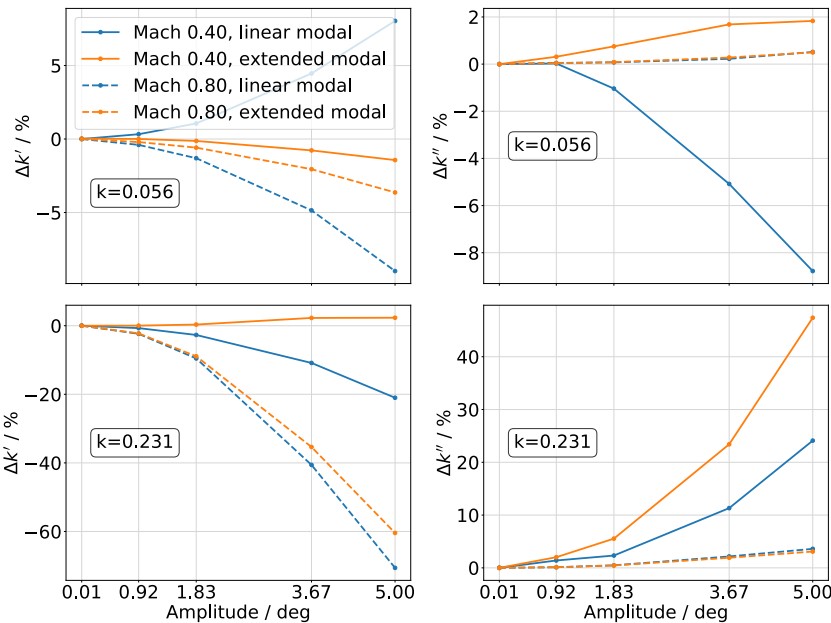

**Figure 15.** Impact of fluid compressibility on aerodynamic stiffness and damping of $Q_{hh}(2,1)$ (viscous flow).

The essential observations regarding the impact of fluid compressibility on the stiffness and damping of the aerodynamic coupling term for viscous flow may be summarized as follows:

- Aerodynamic stiffness:
  - Increased nonlinearity for low and high reduced frequencies;
  - Reduced impact of geometric nonlinearity, especially at high reduced frequencies.
- Aerodynamic damping:
  - Nonlinearity is almost entirely canceled;
  - No impact of geometric nonlinearity.

## 4. Discussion

The results presented above give insight into an aerodynamic nonlinearity for a motion pattern that represents a structural elastic degree of freedom encountered in typical T-tail flutter mechanisms. As this study originates from a generic T-tail with a flutter mechanism close to the reduced frequency of 0.231, the discussion of the physical reasons leading to the aerodynamic nonlinearities and the conceivable impact on T-tail flutter will be limited to the high reduced frequency results.

### 4.1. Physical Sources for Aerodynamic Nonlinearities

Studying the results presented above raises the question for the physical reasons leading to the nonlinear character of the aerodynamic coupling term. As this GAF term is an aerodynamic yaw moment due to HTP roll, there needs to be an asymmetric aerodynamic force induced by the rolling motion that performs mechanical work on the yaw motion and is nonlinear with respect to the displacement amplitude. That is, an asymmetric longitudinal force component, as well as an asymmetric lateral force component are possible sources for the elaborated nonlinearity. Fluid viscosity does not alter the observed aerodynamic nonlinearity significantly; thus, the study of physical reasons leading to the nonlinear work terms is illustrated for viscous flow at Mach 0.4 and high reduced frequency; see Figure 16. Here, the generalized aerodynamic force components are evaluated at the CFD surface node

level and separated into a longitudinal and a lateral component. The positive semi-span visualizes the results based on the linear modal approach and the negative semi-span those based on the extended modal approach. For the illustration, the amplitude-normalized complex aerodynamic responses on lower and upper surface are summed up first for both the smallest, as well as the largest displacement amplitudes. Then, the difference between the largest and smallest displacement amplitude is calculated and projected on the mean camber surface. If the aerodynamic response was linear, the resulting surface value magnitudes would be zero everywhere, which was not the case for the evaluated simulations. Regarding the longitudinal component (Figure 16a), a nonlinearity close to the HTP tips is evident for the linear, as well as the extended modal approach, suggesting the assumption that the tip vortex is a relevant contributor to the aerodynamic nonlinearity. Visually, the extended modal approach results in almost the same location and magnitude of the nonlinearity at the wing tip as the linear modal approach. However, the linear modal approach indicates an additional aerodynamic nonlinearity at the leading edge, which is reduced for the extended modal approach. Figure 16b illustrates the lateral generalized aerodynamic force components, again highlighting a nonlinearity close to the HTP tips. In addition and most notably, the linear modal approach shows a distributed nonlinearity and a distinct difference from the extended modal approach. With the consideration of geometric nonlinearity, this nonlinear aerodynamic work term is largely diminished.

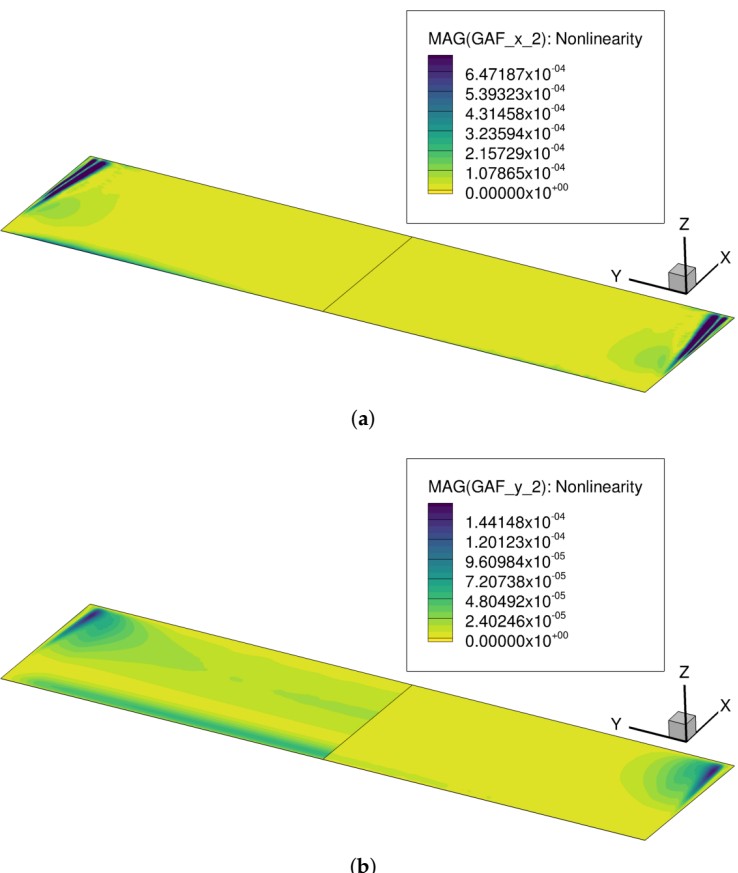

**Figure 16.** Spatial locations of nonlinear components of $Q_{hh}(2,1)$ (viscous flow, Mach 0.4. Positive semi-span: linear modal, negative semi-span: extended modal). (**a**) Longitudinal GAF component; (**b**) lateral GAF component.

These findings are emphasized by evaluating the differences in aerodynamic nonlinearities between the linear and extended modal approach; see Figure 17. One slice in the spanwise direction at $x/c = 0.725$, with $c$ being the chord length, and one slice in the chordwise direction at $y/s = 0.7$, with $s$ being the semi-span, further detail the impact

of the geometric nonlinearities on the aerodynamic nonlinearities in terms of numerical quantities. The GAF difference $\Delta Q_{hh}(2,1)$ illustrates the nonlinearity of the aerodynamic work term for the linear (blue solid line), as well as the extended modal approach (orange solid line), in addition to the difference between the approaches (dark gray solid line). As can be seen in the upper left Figure 17a, the nonlinearity of the longitudinal aerodynamic work term at the wing tip agrees between the linear and extended modal approach, which also holds true for the remaining linear terms along the spanwise slice. At the leading edge, however, the geometric nonlinearity linearizes the longitudinal aerodynamic work term; see upper right Figure 17b. The remaining values along the chordwise slice are rather linear and identical between the linear and extended modal approach. The lateral aerodynamic work terms show a spanwise distributed nonlinearity for the linear displacements, as stated above and shown in the lower left Figure 17c. The second-order displacement terms linearize this GAF element except for at the wing tip, where the nonlinearity remains unchanged. The lower right Figure 17d further illustrates the linearization of the lateral aerodynamic work components at the leading edge when geometric nonlinearities are considered. The remaining chordwise distributed GAF values show a reduced nonlinearity as well when geometric nonlinearities are accounted for. Considering the numerical values for the GAF differences, the lateral components are revealed to be an order of magnitude smaller compared to the longitudinal terms, but with a distributed character rather than a local one. An impact on the integral quantity, thus, cannot be ruled out.

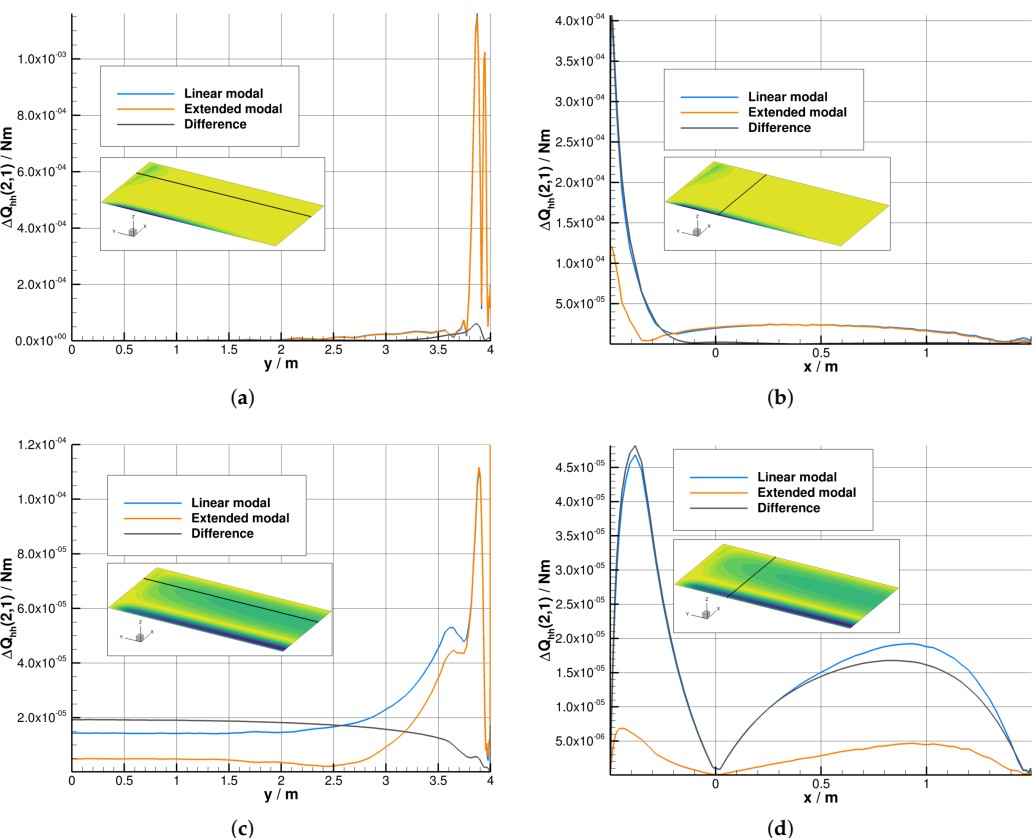

**Figure 17.** Nonlinear components of $Q_{hh}(2,1)$ along spanwise and chordwise slices (viscous flow, Mach 0.4). (**a**) Longitudinal component, spanwise distribution at $x/c = 0.725$; (**b**) longitudinal component, chordwise distribution at $y/s = 0.7$; (**c**) lateral component, spanwise distribution at $x/c = 0.725$; (**d**) lateral component, chordwise distribution at $y/s = 0.7$.

### 4.2. Impact of Fluid Compressibility

Increasing the Mach number from subsonic to transonic speed has been shown to affect the stiffness and damping nonlinearities and to reduce the relevance of geometri-

cally nonlinear displacement terms. Figure 18 illustrates the sources for the aerodynamic nonlinearities for the linear and extended modal approaches. The nonlinearities of the longitudinal and lateral work terms concentrated at the HTP tips are again visible, in addition to a strong contribution of the shock to both terms. Especially the longitudinal component reveals a distributed amplitude-dependent character, which appears to be similar for both modal approaches. Consistent with the observations made for subsonic flow, the lateral term shows deviations between the linear and extended modal approach in terms of the distributed nonlinearity.

The differences between the linear and the extended modal approach are again visualized by two slices, one in the spanwise direction at $x/c = 0.725$ and one in the chordwise direction at $y/s = 0.7$. As is evident, the relative deviations between the linear and the extended modal approach are minor for the longitudinal GAF component (Figure 19a,b) and large for the lateral GAF component (Figure 19c,d, respectively). As the aerodynamic nonlinearities at the shock are only local, a significant contribution to the integral quantity is not expected. On the contrary, the distributed nonlinear terms, which additionally deviate between the linear and the extended modal approach and have already been observed for subsonic flow, are presumably noticeable for the integral quantity.

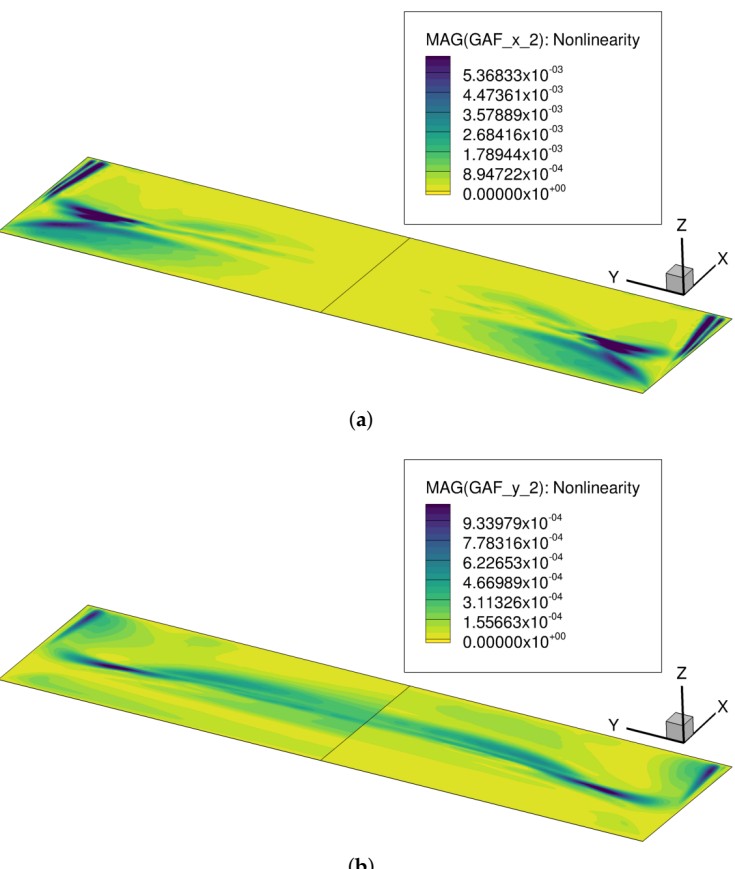

**Figure 18.** Spatial locations of nonlinear components of $Q_{hh}(2,1)$ (viscous flow, Mach 0.8. Positive semi-span: linear modal, negative semi-span: extended modal). (**a**) Longitudinal GAF component; (**b**) lateral GAF component.

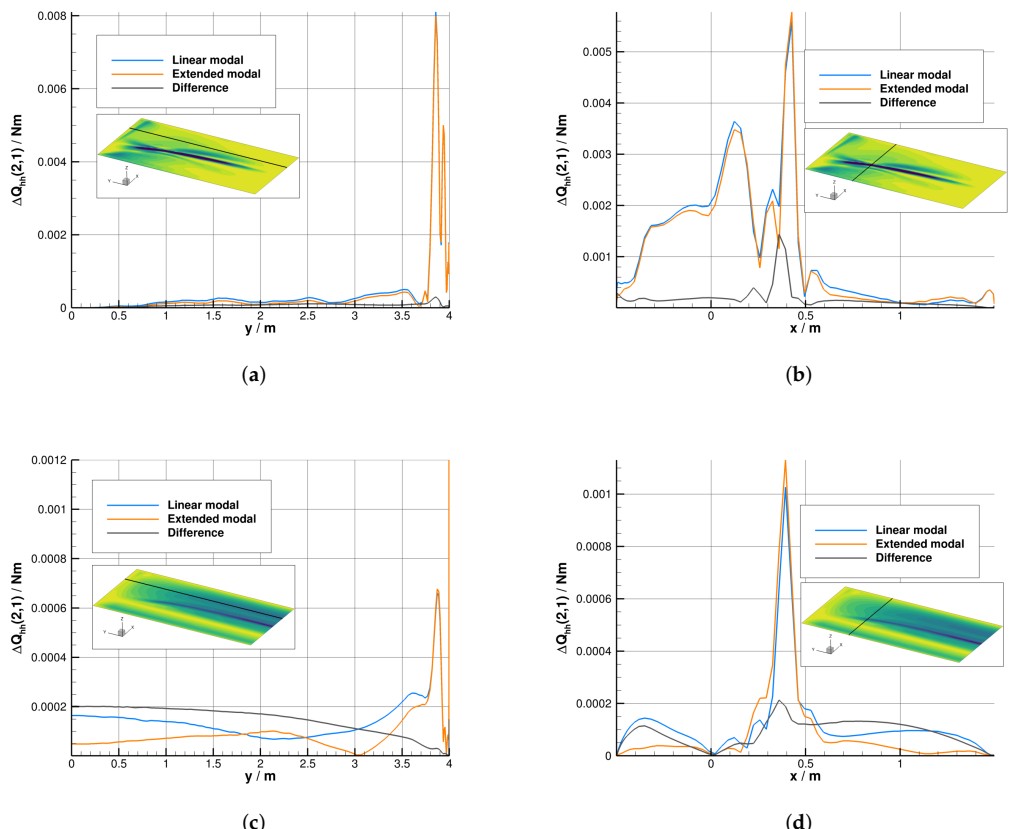

**Figure 19.** Nonlinear components of $Q_{hh}(2,1)$ along spanwise and chordwise slices (viscous flow, Mach 0.8). (**a**) Longitudinal component, spanwise distribution at $x/c = 0.725$; (**b**) longitudinal component, chordwise distribution at $y/s = 0.7$; (**c**) lateral component, spanwise distribution at $x/c = 0.725$; (**d**) lateral component, chordwise distribution at $y/s = 0.7$.

### 4.3. Summary of Aerodynamic Coupling Term Nonlinearity and Implications for T-Tail Flutter

The nonlinearities in aerodynamic stiffness and damping were observed to be of a quadratic kind (cf. Figures 11, 13 and 15). A second-order polynomial was fit through the data, and the leading coefficients resulting from the curve fit were used to quantify the non-linearity. Figure 20 illustrates the low impact of fluid viscosity on the overall stiffness and damping nonlinearity for the subsonic Mach number results. The geometrically nonlinear displacement components resulted in a significant reduction of the stiffness nonlinearity and an increase of the damping nonlinearity for inviscid and viscous flow. At the transonic Mach number, the distinct stiffness nonlinearity in addition to the negligible damping nonlinearity can be recognized. The quadratic displacement components have a slightly mitigating effect on the stiffness nonlinearity and, furthermore, can be neglected for the damping. As the change in stiffness of the aerodynamic coupling term introduced by the quadratic displacement components is dependent on the deformation amplitude, it is insufficient to simply account for the spurious stiffening of the diagonal GAF matrix element and to perform the subsequent amplitude-dependent simulations with a geometrically linear structural model. As the aerodynamic stiffness for the non-autonomous system studied here is observed to be reduced with increasing deformation amplitude, the autonomous system presumably experiences a stiffening nonlinearity, which potentially results in an LCO with instabilities above the linear flutter onset (supercritical or benign LCO) [33]. Here, the linear modal approach might show a smaller LCO amplitude compared to the extended modal approach, as the nonlinearity is shown to be larger when the structural displacements are considered linearly.

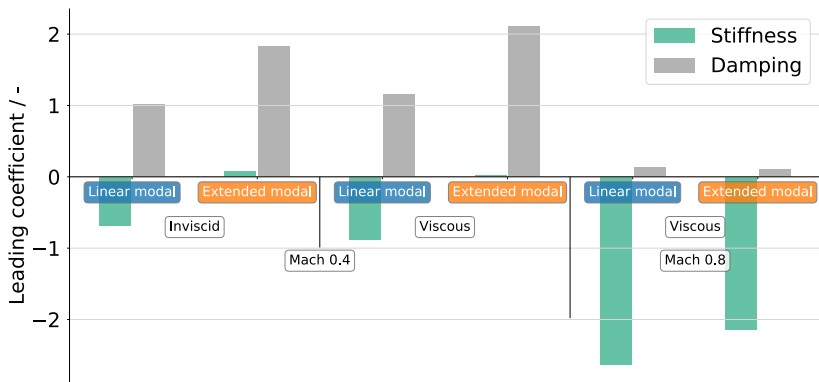

**Figure 20.** Summary of aerodynamic coupling term nonlinearity (reduced frequency 0.231).

## 5. Conclusions and Outlook

An amplitude dependency of aerodynamic forces resulting from HTP roll motion that perform mechanical work on the HTP yaw motion was identified. At a subsonic Mach number, this nonlinearity was shown to be in both stiffness and damping. The geometric nonlinearities showed a strongly mitigating impact on the aerodynamic stiffness nonlinearity and an amplifying impact on the aerodynamic damping nonlinearity. At a transonic Mach number, only the stiffness nonlinearity remained, which was shown to be reduced when accounting for geometric nonlinearities. The reasons for these nonlinearities were identified to be related to the wing tip vortex and, regarding linear displacements, to a contribution by longitudinal forces at the leading edge and one by lateral forces, which are distributed across the entire wing. The latter two sources diminished when taking into account geometric nonlinearities, and only the nonlinear terms close to the wing tips remained, suggesting these as sources of the elaborated aerodynamic stiffness nonlinearity. Fluid viscosity was shown to have a minor impact on the aerodynamic nonlinearity, but led to a stiffness offset, which might affect the linear flutter onset predictions. These findings indicate that it is reasonable to include geometric nonlinearities for amplitude-dependent T-tail flutter studies, as the aerodynamic stiffness nonlinearity was reduced when nonlinear geometric displacement terms were accounted for. For a self-excited system, the linear structural displacement might therefore result in a benign LCO with a smaller amplitude compared to the results based on nonlinear structural displacements.

As the studies have merely given an insight into a nonlinear aerodynamic coupling term, its relevance for T-tail flutter is still speculative. There is a need to address the amplitude-dependent stability of the autonomous system without and with geometric nonlinearities in order to put the suggestions made in this work regarding the impact of the elaborated aerodynamic nonlinearity into perspective. Moreover, previous studies of the author have led to focusing on a positive incidence angle of 3.0°, which is not necessarily a conventional operation point of an HTP. A negative incidence angle resulting in a steady downforce would be suitable to obtain insight into the character of the stiffness nonlinearity at a more realistic operation point. In doing so, the dependency of the nonlinearity on the incidence angle and, with this, on the steady reference state can be addressed. Furthermore, the coupled quadratic mode shape components were, in this particular case of orthogonal rotations, zero. For a more realistic T-tail configuration with sweep and taper, the coupled quadratic mode shape components would be non-zero and have an impact on the aerodynamic coupling term even at the smallest displacement amplitude as well. Sweep and taper would additionally result in a potentially smaller tip vortex and reduce the nonlinearity of the aerodynamic coupling term.

**Funding:** This project has received funding from the Clean Sky 2 Joint Undertaking (JU) under grant agreement No. CS2-LPA-GAM-2020-2023-01. The JU receives support from the European Union's Horizon 2020 research and innovation programme and the Clean Sky 2 JU members other than the Union.

**Institutional Review Board Statement:** Not applicable.

**Informed Consent Statement:** Not applicable.

**Data Availability Statement:** The data presented in this study are available on request from the corresponding author.

**Acknowledgments:** The author would like to express his sincerest gratitude towards Dr. Louw van Zyl from CSIR South Africa and Dr. Markus Ritter from German Aerospace Center (DLR) for their expert advice regarding the computation and verification of quadratic mode shape components.

**Conflicts of Interest:** The author declares no conflict of interest.

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
