# Peer review of "Influence of Fluid Viscosity and Compressibility on Nonlinearities in Generalized Aerodynamic Forces for T-Tail Flutter"

_aerospace, doi:10.3390/aerospace9050256_

Round 1

Reviewer 1 Report

In this paper, the effect of fluid viscosity and compressibility on the nonlinearity of generalized aerodynamic forces for tail flutter is investigated.  Novelty of this paper is appropriate and the Manuscript can be accepted after minor revision:

  • The objective, methodology, and results should be better described, discussed and justified.
  • The results should be expanded significantly and quantitatively.

Reviewer 2 Report

General Comment

The authors analyze the aerodynamic nonlinear effects on the generalized aerodynamic forces that can be used to estimate T-tail flutter. I found the paper interesting and covering a current research topic. However, the paper should be improved, namely section 2, to clarify the work conducted.

Comments and Question

From my point of view, the literature review in the introduction section should be expanded with more references.

I suggest to introduce a paragraph describing how the paper is organized at the end of the introduction section.

Why do you represent the nonlinear physical displacements as a function of the mode shapes?

What is the meaning of the blue and orange colors in Table 1?

Why do you need two CFD models, one for viscous and another for inviscid flow? Why not only one capable of estimating viscous effects?

What are the viscous and inviscid flow models? Are RANS and Euler equations, respectively?

You mention a desired y+ value of 1 according to the literature, although this value will depend on the turbulence model that you have chosen. What was the turbulence model selected for your study?

Did you perform a mesh dependency study for the viscous CFD analyses too?

Details about the wing model should be provided in section 2.

I suggest the introduction of a flowchart summarizing the methodology followed to help the reader understanding it better.
